# Dendritic Cells Internalize *Staphylococcus aureus* More Efficiently than *Staphylococcus epidermidis*, but Do Not Differ in Induction of Antigen-Specific T Cell Proliferation

**DOI:** 10.3390/microorganisms8010019

**Published:** 2019-12-20

**Authors:** Payal P. Balraadjsing, Esther C. de Jong, Willem J. B. van Wamel, Sebastian A. J. Zaat

**Affiliations:** 1Department of Medical Microbiology, Amsterdam Infection and Immunity Institute, Amsterdam UMC, University of Amsterdam, Meibergdreef 15, 1105 AZ Amsterdam, The Netherlands; 2Department of Experimental Immunology, Amsterdam Infection and Immunity Institute, Amsterdam UMC, University of Amsterdam, Meibergdreef 15, 1105 AZ Amsterdam, The Netherlands; e.c.dejong@amsterdamumc.nl; 3Department of Medical Microbiology and Infectious Diseases, Erasmus Medical Center, Wytemaweg 80, 3015 CN Rotterdam, The Netherlands; w.vanwamel@erasmusmc.nl

**Keywords:** *Staphylococcus aureus*, *Staphylococcus epidermidis*, dendritic cells, T cells, superantigen, human immune response

## Abstract

*Staphylococcus aureus* and *Staphylococcus epidermidis* are related species which can cause predominantly acute and subacute infections, respectively. Differences in human adaptive immune responses to these two species are not well understood. Dendritic cells (DCs) have an important role in the control and regulation of anti-staphylococcal T cell responses. Therefore, we aimed to compare the ability of *S. aureus* and *S. epidermidis* to influence the essential steps in human DC activation and subsequent antigen-specific CD4^+^ T cell proliferation, and to investigate the underlying mechanisms. Using multiple strains of both species, we observed that *S. aureus* was internalized more effectively than *S. epidermidis* by DCs but that both species were equally potent in activating these host cells, as evidenced by similar induction of DC maturation marker expression and antigen loading onto MHC-II molecules. The DCs stimulated by *S. aureus* strains not harboring superantigen (SAg) genes or by any of the *S. epidermidis* strains, induced low, likely physiological levels of T cell proliferation. Only DCs stimulated with *S. aureus* strains harboring SAg genes induced high levels of T cell proliferation. Taken together, *S. aureus* and *S. epidermidis* do not differently affect DC activation and ensuing antigen-specific T cell proliferation, unless a strain has the capacity to produce SAgs.

## 1. Introduction

*Staphylococcus aureus* and *Staphylococcus epidermidis* are two major opportunistic pathogens colonizing cutaneous and mucosal surfaces in the human body. Around 30% and 100% of the human population is colonized with *S. aureus* or *S. epidermidis*, respectively [1,2]. In general, these microorganisms have a commensal relationship with the human host. However, when these staphylococci penetrate the epithelial protective barrier in case of trauma or implantation of medical devices, they can become pathogenic. *S. epidermidis* is primarily associated with subacute infections related to any kind of implanted medical device, resulting in a myriad of infections such as catheter-related infections, prosthetic valve endocarditis and implant-associated osteomyelitis [3]. In contrast, the more virulent *S. aureus* is associated with more acute and pyogenic infections ranging from superficial infections to life-threatening invasive diseases such as pneumonia, acute endocarditis, medical device-associated infection and sepsis [4]. 

The successful control and elimination of staphylococci depends on the hosts innate and adaptive immunity. Among cells of the innate immune system dendritic cells (DCs) have a key function in activating adaptive immunity, mostly due to their strategic location at epithelial surfaces and their capacity to acquire, process and present antigens via major histocompatibility complex (MHC) molecules to T cells. Recent studies highlighted the important role of DCs in activating and regulating anti-staphylococcal T cell responses [5,6,7,8]. However, a detailed understanding of the interaction of DCs with *S. aureus* and *S. epidermidis* leading to T cell activation is lacking, and studies have focused particularly on mouse models rather than on human primary cell models. 

In *S. aureus* bloodstream infection in mice, DCs contribute to the control of infection by producing interleukin (IL) 12 [7], a cytokine involved in the development of T helper 1 (Th1) cell responses. Depletion of DCs causes substantial reduction of clearance of bacteria from the lungs and kidneys [7]. *S. aureus* has developed different strategies to evade or modulate DC and T cell responses. They can exacerbate T cell proliferation and pro-inflammatory DC responses in an antigen non-specific manner by producing superantigens (SAgs) which cross-link T cell receptors with MHC class II (MHC-II) molecules on DCs [5,8]. This non-specific stimulation may lead to pathogenic immune responses, as the resulting high concentrations of pro-inflammatory cytokines produced may cause a status of shock possibly followed by death [9]. *S. aureus* can also evade phagocytic killing and persist intracellularly within multiple professional and non-professional phagocytic cells, including mouse DCs [7,10]. 

In contrast to *S. aureus*, *S. epidermidis* lacks the aggressive immune evasion strategies which affect DC and T cell responses. In mouse skin, resident DCs orchestrate T cell responses to commensal *S. epidermidis*, and these responses help to maintain the adaptive immune barrier against invasive microbes [6]. Moreover, *S. epidermidis* or their cell-free supernatants induce DCs to become semi-mature and may cause anti-inflammatory DC responses leading to regulatory T cell induction [5,11]. Such low-grade inflammatory DC and T cell responses may be beneficial in case *S. epidermidis* residing as commensals on the skin, but in case the bacteria cause infection, a more pro-inflammatory response would be required. *S. epidermidis* express poly-γ-glutamic acid (PGA), an extracellular polymer which protects these bacteria from antimicrobial peptides and phagocytic uptake by neutrophils [12,13]. It is not known if PGA also efficiently protects *S. epidermidis* from DC phagocytosis and limits ensuing antigen presentation to T cells. 

It has remained largely unexplored whether viable *S. aureus* and *S. epidermidis* differently affect human DC- and ensuing DC-induced T cell activation, and what the possible mechanisms are underlying such differences. Therefore, to better understand the pathogenesis of *S. aureus* and *S. epidermidis*, we aimed to compare the ability of these bacteria to influence the essential steps in DC activation and subsequent antigen-specific CD4^+^ T cell proliferation. 

## 2. Materials and Methods

### 2.1. Staphylococcal Strains

*S. aureus* strains ATCC 49230, LUH15101 (methicillin-doxycycline resistant) [14], RN4220 (ATCC 35556), JAR060131 [15], 42D (ATCC 27712), and *S. epidermidis* strains O-47 [16], RP62a (methicillin-resistant, ATCC 35984), AMC5 [17] and NCTC100892, were used. The strains selected are clinical isolates (except for strain RN4220), which possess factors important to establish an infection. These strains are often used for in vitro and in vivo studies on staphylococcal pathogenesis, biomaterial-associated infection and treatment with antimicrobial peptides [14,18,19,20,21]. All strains were positive for *icaADBC* gene cluster and all *S. aureus* strains were positive for either the *cap5* or *cap8* gene (data not shown). Prior to each experiment the bacteria were cultured in tryptic soy broth (TSB, BD Difco, Sparks, MD, USA), to the logarithmic growth phase at 37 °C while shaking. The viable bacteria were harvested by centrifugation and resuspended to the desired concentrations in Iscove’s Modified Dulbecco’s Medium (IMDM, Lonza, Basel, Switzerland) containing 10% heat-inactivated (HI) fetal calf serum (FCS, Invitrogen, Carlsbad, CA, USA). Cell-free culture supernatants of *S. aureus* strains were obtained by centrifugation followed by passing the supernatants through 0.2 μm filters. Absence of viable bacteria in the supernatant was confirmed by culture on blood agar plates. For internalization experiments GFP-expressing *S. aureus* ATCC 49230 [18] and *S. epidermidis* O-47 [20,22] or carboxyfluorescein succinimidyl ester (CFSE)-labeled staphylococci were used. In brief, GFP-staphylococci were obtained by transformation with plasmid WVW189 containing the *gfp_uvr_* gene, as previously described [18,20]. The fluorescent intensity of GFP-expressing *S. aureus* ATCC 49230 and *S. epidermidis* O-47 was the same (data not shown). CFSE-staphylococci were labeled by incubation in 0.5 µM CFSE (Invitrogen) for 30 minutes at room temperature followed by two washing steps.

### 2.2. Generation and Stimulation of DCs

Human peripheral blood was collected after obtaining written informed consent in accordance with the approval of the Medical Ethical Committee of the Amsterdam UMC, Location AMC, Amsterdam. Monocyte-derived DCs were generated and cultured from peripheral blood of anonymous healthy human blood donors as previously described [23]. In brief, monocytes were isolated by density centrifugation on Lymphoprep (Nycomed, Zürich, Switserland) and Percoll (GE Healthcare, Chicago, IL, USA) and monocytes (4 × 10^5^ cells/mL) were cultured for 6 days in 24-well culture plates (Costar, Cambridge, MA, USA) in IMDM (Thermo Fisher Scientific, Waltham, MA, USA) supplemented with 5% FCS, 86 µg/mL gentamicin (Duchefa, Haarlem, The Netherlands), 500 U/mL recombinant human GM-CSF (Schering-Plough, Kenilworth, NJ, USA) and 10 IU/mL recombinant human IL-4 (Miltenyi Biotec, Bergisch Gladbach, Germany) to obtain DCs. The yield of monocytes using the two-step density centrifugation cell isolation method on Lymphoprep and Percoll is 75%–90%. On day 6, immature DCs (iDCs, CD11c^+^CD14^−^
Appendix A) were resuspended and stimulated for the indicated times with the bacteria in cell culture medium (IMDM supplemented with 10% HI FCS and 86 µg/mL gentamicin, unless indicated otherwise). The multiplicity of infection (MOI) used in these co-culture experiments was 100, 50 or 20 colony forming units (CFU) per DC. No major differences in DC viability nor in the proportion of apoptotic cells were observed after stimulation, as examined by flow cytometry using propidium iodide (Sigma, St Louis, MO, USA) and Annexin V (BD Biosciences, Franklin Lakes, NJ, USA) staining, respectively (Appendix A).

### 2.3. DC Maturation Analysis

DCs (1 × 10^5^ cells) were stimulated with staphylococci at an MOI of 100 in 1 mL cell culture medium for 48 h. To prevent bacterial overgrowth, 10 µg/mL moxifloxacin (Avelox, Bayer Schering Pharma, West Haven, CT, USA) was added. As measure of DC activation the expression of cell surface molecules was measured by flow cytometry (Canto II, BD Biosciences) after incubation with fluorescent antibodies HLA-DR-PerCP, CD83-APC and CD86-PE (all purchased from BD Biosciences). For each fluorescent antibody, cells stained with a single fluorophore were used to correct for false positive fluorescence (data not shown). Lipopolysaccharide (LPS) (100 ng/mL from *Escherichia coli* 0111:B4; Sigma-Aldrich, St Louis, MO, USA) was used as positive control for the induction of fully mature DCs.

### 2.4. Internalization and Processing Assays

iDCs (2 × 10^4^ cells) were co-cultured with GFP-expressing or CFSE-labelled staphylococci (37 °C, 20 CFU/ cell) in 200 µL cell culture medium for 2, 4, 8 or 24 h. Uptake was stopped by washing three times with ice-cold PBS and cells were analyzed by flow cytometry. Antibodies against lipoteichoic acid (LTA, QED Bioscience Inc., San Diego, CA, USA) followed by secondary fluorescent GαM-PE (Jackson ImmunoResearch, West Grove, PA, USA) were used to detect extracellular bacteria. DCs with a single GFP stain were used to allow proper gating of GFP-positive and α-LTA-PE-negative cells (data not shown). For confocal laser scanning microscopy (CLSM), iDCs and GFP-expressing bacteria were incubated on poly-d-lysine-coated (Sigma Aldrich) coverslips for 1 h at 37 °C. After incubation, cells were washed, fixed with 3.7% paraformaldehyde (Sigma-Aldrich) and stained with primary anti-CD11c (clone B-ly6, BD Pharmingen, San Diego, CA, USA) followed by GαM-Alexa-568 (Molecular Probes, Leiden, The Netherlands), primary anti-LTA followed by GαM-Alexa-700 (Molecular Probes, Leiden, The Netherlands) and Hoechst (Immunochemistry Technologies, Bloomington, MN, USA) to visualize the DC membrane, extracellular bacteria and DNA, respectively. Cells were analyzed with a confocal microscope (Leica SP8 X, Leica LAS-X software, Wetzlar, Germany). To investigate antigen processing, iDCs (1 × 10^5^ cells) were co-cultured with staphylococci at an MOI of 50 or LPS (100 ng/mL) as positive control for 17 h in cell culture medium. DCs were stained with anti-HLA-DR-APC (clone L243, BD Bioscience) and anti-CerCLIP.1-FITC (BD Bioscience), to detect human CLIP bound to HLA-DR, and subsequently analyzed by flow cytometry. To calculate the relative expression of CLIP on DCs, the geometric mean fluorescent intensity (gMFI) of CerCLIP was divided by the gMFI of HLA-DR.

### 2.5. Intracellular Bacterial Survival

iDCs (2 × 10^4^ cells) were allowed to internalize staphylococci (MOI of 20) for 1 h at 37 °C in cell culture medium without antibiotics in a 96-well plate. After 1 h, DCs were washed for 10 min with culture medium containing 86 µg/mL gentamicin to kill the remaining extracellular bacteria. DCs infected with *S. epidermidis* strain RP62a were washed for 10 min with culture medium containing 10 µg/mL moxifloxacin since this strain is resistant against gentamicin. DCs were either harvested to obtain the number of internalized bacteria after 1 h or further incubated at 37 °C for 3, 7, 23 and 47 h in cell culture medium containing 10 µg/mL gentamicin or 1 µg/mL moxifloxacin for DCs with *S. epidermidis* RP62a. The gentamicin dose (10 µg/mL) used during the different incubation periods is not likely to affect the intracellular bacteria numbers [24]. At each sampling time point harvested DCs were washed with cell culture medium without antibiotics and lysed with 0.1% Triton X-100 in phosphate buffered saline (PBS). The DC lysate was serially diluted and plated on blood agar plates. After 24 h of incubation at 37 °C, the numbers of viable bacteria were quantified.

### 2.6. T Cell Proliferation

Human peripheral blood lymphocytes (PBLs) were isolated as previously described [25]. Autologous or allogeneic naive CD4^+^ T cells were isolated from PBLs by negative selection using the CD4^+^ T cell isolation MACS kit (Miltenyi Biotec, Bergisch Gladbach, Germany) followed by CD45RO-PE (Dako, Agilent, Santa Clara, CA, USA) and magnetic anti-PE beads (Miltenyi-Biotech, Bergisch Gladbach, Germany), as described previously [25]. The purity of the isolated naive CD4^+^ (CD45RO^-^CD45RA^+^) T cells was >97%, as assessed by flow cytometry. DCs were stimulated for 48 h with viable bacteria as described for the DC maturation analysis, washed, and co-cultured with CFSE-labeled (0.5 µM, Invitrogen) autologous naive CD4^+^ T cells (CD45RO^-^CD45RA^+^) at a 1:1 ratio (4 × 10^4^: 4 × 10^4^ cells) for 5 days in cell culture medium. To assess the induction of antigen-independent T cell proliferation by soluble *S. aureus* or *S. epidermidis* factors, DCs were stimulated for 48 h with supernatants of *S. aureus* or *S. epidermidis* cultures (diluted 1:5) and co-cultured with CFSE-labeled allogeneic naive CD4^+^ T cells, instead of autologous T cells. Sterile TSB bacterial culture medium was used as control. T cell proliferation was determined by flow cytometry (Canto II, BD Biosciences) and quantified using FlowJo software (version 7.6.5, Tree star, Ashland, OR, USA). The precursor frequency indicates the percentage of cells of the original T cell population that underwent at least one cell division.

### 2.7. Staphylococcal Cap Locus PCR and Superantigen Multiplex PCR

DNA was extracted from the bacteria and PCRs and multiplex PCRs were performed to assess the presence of *capB*, *capC*, *capA* and *capD* genes and 19 different staphylococcal enterotoxins (broadly classified as superantigens) genes (for primers see Table 1), respectively. Multiplex PCRs were performed as previously described [26,27]. In brief, extracted DNA was used in five sets of multiplex PCRs targeting the following sets of genes: (i) *sea*, *seh*, *sec*, and *tst*; (ii) *sed*, and *sek*; (iii) *see*, *seb*, *sem*, *sel*, and *seo*; (iv) *sen*, *seg*, *seq*, and *sej*; (v) *sei*, *ser*, *seu*, and *sep*. The PCR products were resolved by electrophoresis in 1.5% agarose gels (1× Tris–borate–EDTA buffer), stained with ethidium bromide, and visualized under ultraviolet light.

### 2.8. Statistical Analysis

Data were analyzed for statistical significance using linear mixed models on rank transformed data followed by the post hoc Wilcoxon signed ranked test for pairwise comparisons. *p*-values of ≤0.05 were considered as statistically significant. Statistical analysis was performed using IBM SPSS Statistics software version 24.

## 3. Results

### 3.1. S. aureus and S. epidermidis Induce Similar Expression of DC Maturation Markers

Upon the encounter of microbes, DCs will undergo a specific activation program. As the levels of DC activation determine T cell proliferation, we analyzed to what extent *S. aureus* and *S. epidermidis* induce DC activation. DC activation was determined by measuring the expression of DC maturation markers HLA-DR, CD86 and CD83. *S. aureus* and *S. epidermidis* were equally potent in upregulating the expression of these DC maturation markers of which the upregulation of CD86 expression was the most pronounced (Figure 1A). The ability of the bacteria to induce DC activation was not strain specific, as shown by similar expression of DC maturation marker CD86 upon stimulation with different *S. aureus* and *S. epidermidis* strains (Figure 1B). The expression of HLA-DR and CD83 was also similarly upregulated by the different *S. aureus* and *S. epidermidis* strains (Appendix A). These results demonstrate that *S. aureus* and *S. epidermidis* strains induce similar DC maturation.

### 3.2. S. epidermidis Are Internalized by DCs to a Lower Extent than S. aureus

Subsequently, we investigated the capacity of DCs to internalize *S. aureus* and *S. epidermidis*, by incubating DCs with GFP-expressing staphylococci and quantifying internalization by flow cytometry. Interestingly, we found that DCs were more potent in internalizing *S. aureus* than *S. epidermidis* at all time points tested (Figure 2A). The percentage of DCs that internalized *S. aureus* on average was five times higher than the percentage of DCs that internalized *S. epidermidis* (Figure 2A). Moreover, also the mean numbers of *S. aureus* taken up per cell were higher, as indicated by a higher GFP mean fluorescence intensity of DCs incubated with these bacteria (Appendix A). These findings were confirmed by confocal laser scanning microscopy (CLSM), where clearly higher numbers of GFP-expressing *S. aureus* bacteria were detected inside DCs (Figure 2B). All *S. aureus* strains tested were internalized to a higher level than the *S. epidermidis* strains (Figure 2C). Even after a time period of 24 h *S. epidermidis* bacteria were not internalized to the same level as *S. aureus* bacteria, showing that it did not simply require more time to internalize *S. epidermidis* bacteria (Appendix A). Collectively these data indicate that DCs are more efficient in internalizing *S. aureus* bacteria than in internalizing *S. epidermidis* bacteria.

It has been reported that *S. epidermidis* PGA is involved in resistance to neutrophil phagocytosis [13]. The genes of the *cap* locus (*capBCAD*) code for the production of the *S. epidermidis* PGA capsule [12]. We therefore investigated the presence of *capBCAD* genes in the tested *S. aureus* and *S. epidermidis* strains to investigate whether this was associated with the decreased capacity of DCs to internalize *S. epidermidis*. Indeed, the *S. epidermidis* strains were all positive for the *capBCAD* genes, whereas *S. aureus* strains were all negative (Appendix A). This suggests that PGA may play a role in resistance of *S. epidermidis* to DC phagocytosis.

Since *S. aureus* is described to persist within phagocytic cells, we hypothesized that the high level of internalization of *S. aureus* by DCs may result in intracellular survival and multiplication. We therefore infected DCs with *S. aureus* and *S. epidermidis* and lysed the DCs at specific time points post phagocytosis to quantify the numbers of viable internalized bacteria. The intracellular bacterial load in DCs decreased over time for both bacteria, though the reduction in number of intracellular *S. epidermidis* was less than of *S. aureus*. No complete killing of *S. aureus* and *S. epidermidis* bacteria was observed after 48 h, with the exception of *S. aureus* strain 42D (Figure 3).

### 3.3. S. aureus and S. epidermidis Antigens Are both Efficiently Loaded on MHC-II Molecules by DCs

After internalization of bacteria, the ability of DCs to process and load antigens on MHC-II molecules is essential for antigen-specific CD4^+^ T cell activation. A pivotal step in antigen loading on MHC-II molecules is the replacement of class II-associated invariant chain peptide (CLIP) by antigenic peptides in the MHC-II peptide binding groove. High cell surface expression of CLIP is an indicator for low effectiveness of antigen presentation [28]. Conversely, loading of antigenic peptides in the MHC-II binding groove reduces the surface expression of CLIP. To investigate whether the different capacity of DCs to internalize *S. aureus* and *S. epidermidis* may lead to differences in antigen loading on MHC-II molecules, we analyzed the efficacy of CLIP exchange for antigenic peptides by measuring total HLA-DR expression and relative CLIP expression (relative amount of HLA-DR occupied by CLIP) by flow cytometry. In line with previous reports [29,30], stimulation with LPS, which is not processed as antigen on MHC-II molecules, leads to high relative CLIP expression on DCs whereas unstimulated DCs have low relative CLIP expression (Figure 4). Incubation with the different *S. aureus* and *S. epidermidis* strains induced lower relative CLIP expression compared to incubation with the LPS control. Moreover, no differences were found in relative CLIP expression on DCs incubated with the different *S. aureus* and *S. epidermidis* strains (Figure 4). This indicated that despite difference in uptake of the bacteria, DCs did not differently process and present antigenic peptides upon incubation with *S. aureus* or *S. epidermidis* strains.

### 3.4. S. aureus and S. epidermidis Strains Vary in Their Capacity to Induce T Cell Proliferation

Next, we analyzed the ability of DCs having phagocytosed *S. aureus* or *S. epidermidis* to activate antigen-specific CD4^+^ T cells. DCs were incubated with *S. aureus* or *S. epidermidis* for 48 h, washed and then co-cultured with CFSE-labeled autologous naive CD4^+^ T cells. After 5 days of co-culture the precursor frequency was assessed as a marker of T cell proliferation. Clearly, *S. epidermidis*-stimulated DCs were less potent than *S. aureus*-stimulated DCs in inducing proliferation of naive T cells (Figure 5A). Surprisingly, individual *S. aureus* strains differed in their capacity to induce DC-mediated T cell proliferation while all *S. epidermidis* strains consistently induced low levels of T cell proliferation (Figure 5B,C). These data indicated that the T cell stimulatory capacity of *S. aureus*-stimulated DCs was strain specific, whereas for *S. epidermidis*–stimulated DCs this capacity was low for all strains tested.

### 3.5. In Absence of SAgs, S. aureus and S. epidermidis Induce Similar Levels of T Cell Proliferation

All *S. aureus* strains were efficiently phagocytosed and no differences in antigen loading in MHC-II was detected among strains, but still a major difference in T cell stimulatory capacity was observed when DCs were incubated with different strains. The high proliferative responses of T cells to certain strains of *S. aureus* can probably be explained by the production of staphylococcal superantigens (SAgs) by these strains. The difference in potency in inducing T cell proliferation may be related to differences in expression and repertoire of SAgs. In order to investigate a possible role of SAgs in the proliferative responses of autologous T cells we assessed the presence of SAg genes in staphylococcal strains by PCR. All *S. aureus* strains that supported high T cell proliferation were positive for three or more SAg genes. In contrast, all *S. aureus* strains which lacked the capacity to induce high T cell proliferation were negative for any of the SAg genes analyzed (Figure 6A). As expected, the *S. epidermidis* strains were negative for the SAg genes as well. In agreement with the putative role of SAgs in the observed T cell proliferation, exposure of allogeneic T cells to DCs stimulated with bacterial cell-free supernatants of cultures of the SAg-positive strains induced proliferation, whereas supernatants of SAg-negative *S. aureus* strains or *S. epidermidis* strains did not (Figure 6B). Moreover, addition of cell-free supernatant of an SAg-positive *S. aureus* strain increased the levels of T cell proliferation in response to an SAg-negative strain to the same level as induced by the SAg-positive strain itself (data not shown). This confirmed that secreted factors of the *S. aureus* strains positive for SAg genes were responsible for the high T cell proliferation in an antigen non-specific manner, suggesting that these factors were the respective SAgs. Of note, there seemed to be a positive correlation between the number of SAg genes present in *S. aureus* strains and the level of T cell proliferation induced (Figure 6B).

## 4. Discussion

In this study we provide a detailed understanding of the interaction of *S. aureus* and *S. epidermidis* with human dendritic cell leading to T cell proliferation. We show that DCs more effectively internalized *S. aureus* than *S. epidermidis*, but that both bacterial species were equally potent in activating DCs as evidenced by similar induction of DC maturation marker expression and antigen loading on MHC-II molecules. Only certain *S. aureus* strains induced high levels of T cell proliferation, owing to their capacity of secreting superantigens (SAgs). *S. aureus* strains lacking SAg genes induced similar T cell proliferation as *S. epidermidis* strains. Taken together, these observations indicate that the difference in T cell proliferation in response to *S. aureus* and *S. epidermidis* is not due to differences in DC activation, bacterial uptake or antigen processing and presentation but likely to the capacity of *S. aureus* strains to produce SAgs.

It is well known that *S. aureus* produce a wide array of virulence factors that interfere with normal immune function, among which are SAgs inducing high, often pathogenic levels of antigen-independent T cell proliferation and pro-inflammatory cytokine production related to sepsis [9]. However, the presence of SAgs not always is directly correlated with the severity of infection or outcome since invasive clinical *S. aureus* isolates are not always positive for SAgs [31,32]. For example, the SAg gene profiles of invasive and nasal *S. aureus* isolates were shown to be very similar [26]. Three of the five *S. aureus* strains used in this study caused excessive DC-mediated T cell proliferation. These strains carried three or more of 19 analyzed SAg genes. All *S. epidermidis* strains were negative for all SAg genes tested, which is in line with the general consensus that human *S. epidermidis* isolates are negative for SAg genes [33]. Correspondingly, the most potent inducer of T cell proliferation, *S. aureus* LUH15101 was positive for eight different SAg genes. The *S. aureus* strains harboring multiple SAg genes activated larger numbers of T cells than strains producing only a single or a few SAg, corresponding to results with other sets of *S aureus* strains [34,35]. This presumably is the case because the multiple SAgs activate multiple T cell populations expressing distinct Vβ–T cell receptor regions. We did not quantify the level of the specific SAgs present in the supernatant of the SAg-positive *S. aureus* strains, but only minute amounts (pg/ml) of SAgs are needed for human DCs to activate T cells [36,37]. Of note, as in many other studies [38,39,40,41], we base our conclusions on the presence of SAg genes. Assuming that these genes are indeed expressed and that (activating levels of) SAgs are produced. We tested our staphylococcal strains only for the presence of 19 of the 26 known SAg genes [42,43,44,45], because of the lack of positive control strains for the remaining seven genes. Therefore, we cannot exclude the presence of other SAg genes in our strains.

SAg-negative *S. aureus* and the *S. epidermidis* strains were equally potent in inducing DC-mediated T cell proliferation, inducing approximately 5% of the T cells to proliferate. This is in line with published proliferation frequencies of human CD4^+^ naive and memory T cells reactive to such strains or their antigens, which varies between 0.2%–10% [5,46,47]. The variation in this frequency is depending on the human donor’s T cell receptor repertoire, prior exposures to staphylococci and on the staphylococcal antigens used in the experiments. Since the level of naive CD4^+^ T cell proliferation as observed in our study is within the normal range, we assume that this is a physiologically functional level of response to *S. aureus* and *S. epidermidis*. Of note, our proliferation data do not necessarily indicate that *S. aureus* and *S. epidermidis* induce the same Th cell polarization [5,6]. This polarization will depend on the T cell polarizing cytokines produced by DCs upon *S. aureus* or *S. epidermidis* binding to potentially different (combinations of) pathogen recognition receptors.

DCs stimulated with *S. aureus* or *S. epidermidis* equally upregulated the expression of DC maturation markers. In contrast, a study by Laborel-Préneron et al. reported that the cell-free supernatant of *S. aureus* induced high levels of DC maturation marker expression but cell-free supernatant of *S. epidermidis* did not [5]. Apparently, DCs respond weakly to the components present in *S. epidermidis* supernatant while whole viable *S. epidermidis* bacterial cells induced strong DC maturation marker expression. In addition, the high DC maturation marker expression induced by *S. aureus* supernatant is indicated to be largely due to SAgs [5,48,49,50,51,52,53].

DCs internalized significantly higher numbers of *S. aureus* than of *S. epidermidis*, for all strains tested. Although from the confocal images it appeared that adherence of *S. epidermidis* to DCs was impaired, we did observe adherence of *S. epidermidis* by flow cytometry analysis (data not shown). The limited capacity of DCs to internalize *S. epidermidis* was likely due to the presence of the *S. epidermidis* PGA capsule. It is thought that the PGA capsule has a biological role in both the non-infectious and infectious lifestyle of *S. epidermidis*. PGA protects *S. epidermidis* against environmental factors such as high salt concentrations and antimicrobial peptides on the skin, but also helps evading the immune response by preventing phagocytosis [12]. Deletion of the *cap* genes required for synthesis of the *S. epidermidis* PGA capsule, causes an increase of *S. epidermidis* internalization by neutrophils [12]. Other capsular polysaccharides, such as poly-N-acetyl glucosamine (PNAG) also known as polysaccharide intercellular adhesin (PIA) or capsular polysaccharides (CP)5 and CP8, may also protect staphylococci against phagocytosis [54,55,56,57,58]. Since all strains are positive for the for the (*icaADBC*, *cap5*, *cap8*) genes necessary for PNAG/PIA, CP5 and CP8 synthesis, it is unlikely that these capsules are the distinguishing factor in the different level of *S. aureus* and *S. epidermidis* internalization by DCs. In contrast to *S. epidermidis*, which evade the immune response by limiting phagocytosis, *S. aureus* evade extracellular immune responses by actively inducing their internalization through binding of fibronectin and its subsequent recognition by α5β1 integrins on both professional and non-professional phagocytic cells [59,60,61]. This suggests that in addition to *S. epidermidis* expression of anti-phagocytic PGA, expression of pro-phagocytic factors by *S. aureus* may explain the difference in level of internalization of *S. aureus* and *S. epidermidis* by DCs.

There is substantial evidence that *S. aureus* can survive and even multiply in professional phagocytes, including human and murine neutrophils and macrophages [62,63,64,65], however survival in the host is dependent on the MOI and the bacterial growth phase [66,67,68]. *S. epidermidis* bacteria are killed by phagocytic cells, but PIA, a factor involved in *S. epidermidis* biofilm formation, may decrease the intracellular killing [54,69,70,71]. We observed a decrease of numbers of viable intracellular *S. aureus* and *S. epidermidis* in DCs over time, indicating that DCs killed the internalized bacteria of both species. The reduction of numbers of viable intracellular *S. aureus* was stronger than of *S. epidermidis*. In accordance with previous findings, the rate of intracellular killing seemed proportional to the number of internalized bacteria [72]. Although the capacity of DCs to kill the internalized staphylococci was limited, we did not observe any intracellular net growth of the bacteria. In line with our findings, previous studies demonstrated that human DCs have a low efficiency of killing internalized pathogens, especially when compared to human monocytes and macrophages [73,74,75]. This result is in agreement with the main function of DCs, which is to sense and process pathogens and present their antigens to T cells, rather than to eliminate all pathogens. Moreover, we previously reported that DCs which internalized staphylococci undergo the main steps in the process of antigen presentation on MHC-II molecules and actually induce staphylococcal-specific T cell activation [76]. Here we showed that despite differences in bacterial uptake of *S. aureus* and *S. epidermidis*, DCs which had internalized these bacteria were equally efficient in antigen loading onto MHC-II molecules, as indirectly measured by their efficacy of CLIP replacement by antigens. This indicates that *S. aureus* and *S. epidermidis* bacteria are not only internalized and killed by DCs but also processed via the endosome-lysosome pathway leading to staphylococcal-antigen presentation to T cells.

Our findings indicate that despite differences in *S. aureus* and *S. epidermidis* internalization by DCs, the DCs were equally effective in expressing activation markers and antigen presentation through their MHC-II molecules. Differences in the level of the ensuing DC-induced T cell proliferation were attributed to the capacity of *S. aureus* bacteria to produce SAgs. *S. epidermidis* strains and *S. aureus* strains not producing SAgs induced similar levels of T cell proliferation. Thus, *S. aureus* and *S. epidermidis* do not differently affect DC activation and ensuing antigen-specific T cell proliferation in the absence of SAgs.

## Figures and Tables

**Figure 1 microorganisms-08-00019-f001:**
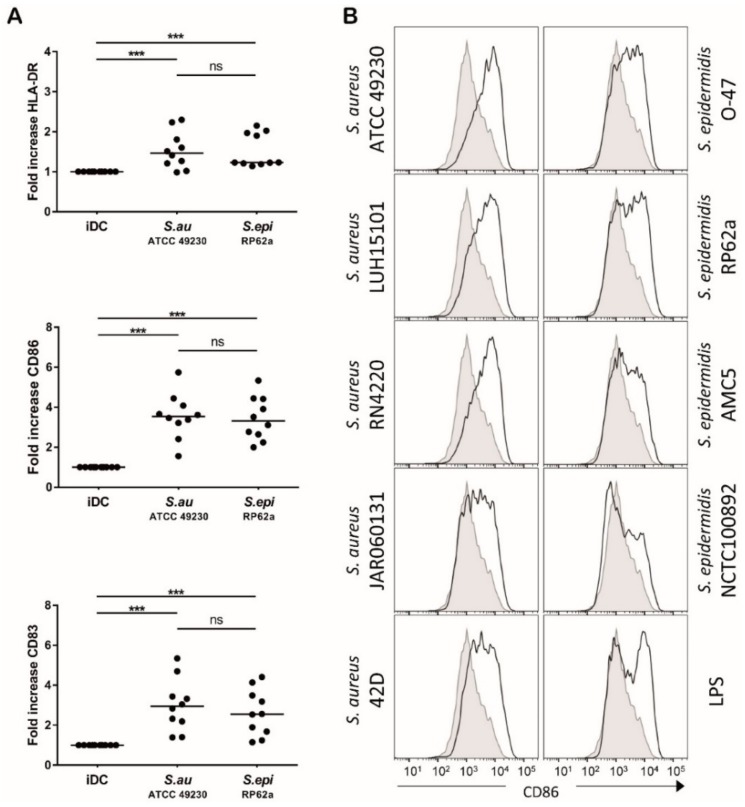
Dendritic cell (DC) maturation marker expression upon *S. aureus* or *S. epidermidis* stimulation. (**A**) Surface expression of HLA-DR, CD86 and CD83 after 48 h on unstimulated DCs (iDCs) or on DCs stimulated with *S. aureus* ATCC49230 or *S. epidermidis* RP62a, as measured by flow cytometry. Fold difference of geometric mean fluorescence intensity of stimulated relative to unstimulated DCs (iDCs). Data of 10 independent experiments. Each dot represents one donor tested in an individual experiment, the horizontal line represents median value, *** *p* < 0.001. (**B**) Expression of CD86 (^10^log fluorescence intensity) on iDCs (filled) or on DCs stimulated with different strains of *S. aureus* and *S. epidermidis* (solid). Data from one experiment out of two performed, with similar results.

**Figure 2 microorganisms-08-00019-f002:**
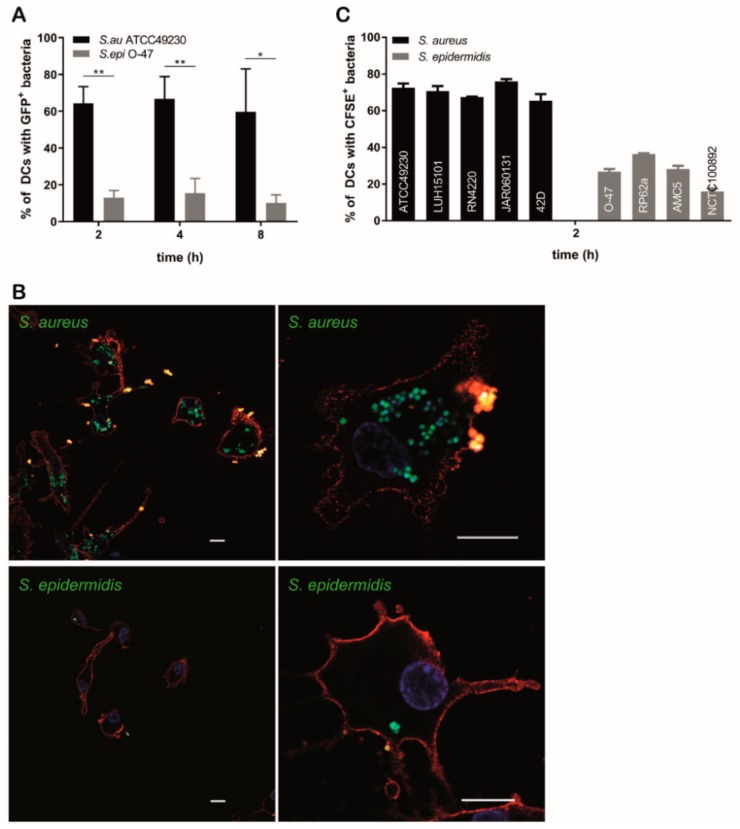
Internalization of *S. aureus* and *S. epidermidis* by DCs. (**A**,**B**) DC uptake of GFP-*S. aureus* ATCC49230 or GFP-*S. epidermidis* O-47. (**A**) Percentages of DCs which internalized fluorescent staphylococci were quantified by flow cytometry. Data are presented as mean + SD of five independent experiments, * *p* < 0.05, ** *p* < 0.01. (**B**) Representative confocal laser scanning microscopy (CLSM) pictures of internalization of GFP-*S. aureus* ATCC49230 (upper panels) and GFP-*S. epidermidis* O-47 (lower pannels) by DCs. DCs were stained with CD11c (red) to visualize cell membranes, hoechst (blue) to visualize DNA and anti-LTA (yellow) was used to discriminate between internalized (not stained) and extracellularly (stained) attached bacteria. The white scale bar indicates 10 µm. (**C**) DC uptake of carboxyfluorescein succinimidyl ester (CFSE)-labeled strains of *S. aureus* or *S. epidermidis* quantified by flow cytometry. Data are presented as mean + SD of duplicate values of one experiment.

**Figure 3 microorganisms-08-00019-f003:**
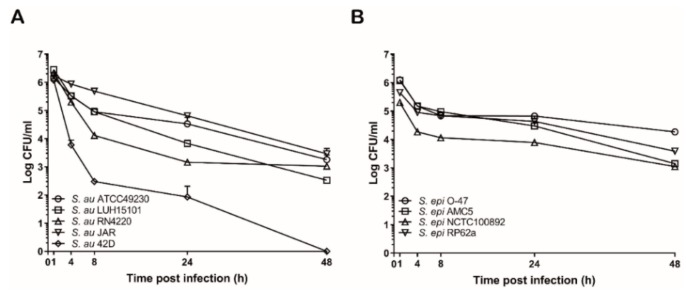
Numbers of viable intracellular *S. aureus* and *S. epidermidis* after phagocytosis in DCs over time. DCs were allowed to internalize different strains of *S. aureus* or *S. epidermidis*. After 1 h of internalization DCs were washed with medium containing 86 µg/mL gentamicin to kill extracellular bacteria and either harvested or further cultured in medium with 10 µg/mL gentamicin (to prevent extracellular growth) for different periods. The intracellular numbers of *S. aureus* (**A**) or *S. epidermidis* (**B**) bacteria were quantified by lysing DCs and quantitative bacterial culture of the DC lysates. Data are presented as mean + SD of duplicate values of one representative experiment out of three.

**Figure 4 microorganisms-08-00019-f004:**
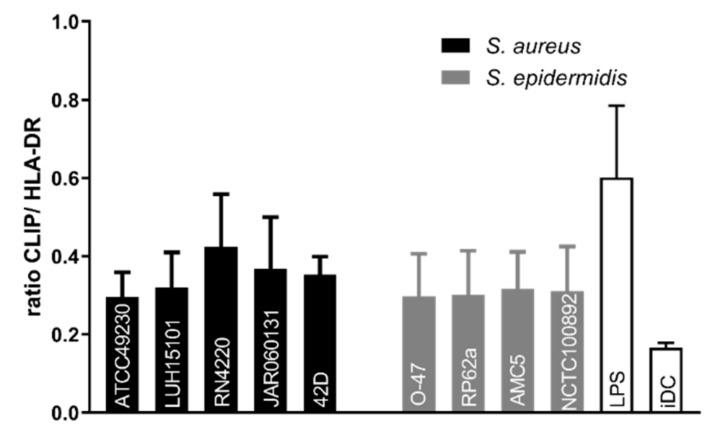
DC antigen loading onto MHC-II molecules. The relative class II-associated invariant chain peptide (CLIP) amount per HLA-DR molecule (CLIP:HLA-DR ratio) on unstimulated DCs (iDC) or DCs incubated with LPS or with different strains of *S. aureus* or *S. epidermidis* for 17 h. Data are presented as mean + SEM of five independent experiments with DCs of individual donors.

**Figure 5 microorganisms-08-00019-f005:**
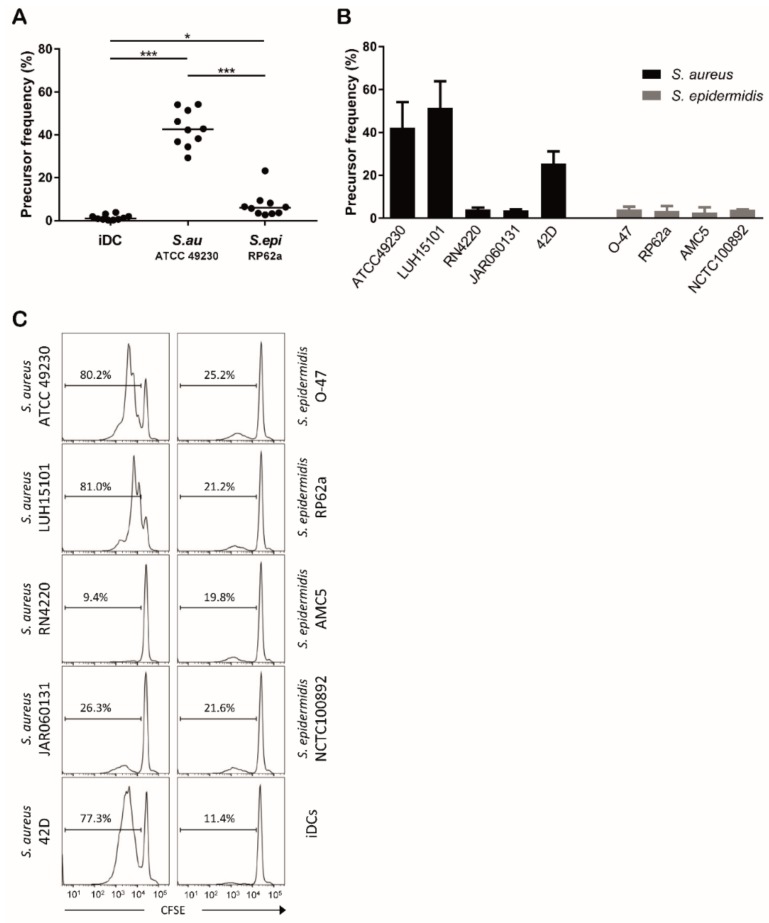
Induction of T cell proliferation by DCs stimulated with *S. aureus* and *S. epidermidis* strains. (**A,B,C**) Proliferation of CFSE labeled naïve CD4^+^ T cells upon co-culture with unstimulated DCs (iDCs), or *S. aureus* or *S. epidermidis* stimulated DCs. (**A**) The percentage of T cells that divided (precursor frequency) upon stimulation with *S. aureus* ATCC49230 or *S. epidermidis* RP62a. Data of 10 independent experiments. Each dot represents one donor tested in an individual experiment, horizontal line represents median value, * *p* < 0.05, *** *p* < 0.001. (**B**) The precursor frequency of T cells upon stimulation by DCs incubated with different strains of *S. aureus* and *S. epidermidis*. Data shown are median + range of three independent experiments with cells of different donors. (**C**) The CFSE profiles of proliferated T cells stimulated by DCs incubated with different strains of *S. aureus* and *S. epidermidis* or unstimulated (iDCs) (^10^log fluorescence intensity). Data from one representative experiment out of the three experiments shown in (**B**). The percentage of proliferated cells is indicated.

**Figure 6 microorganisms-08-00019-f006:**
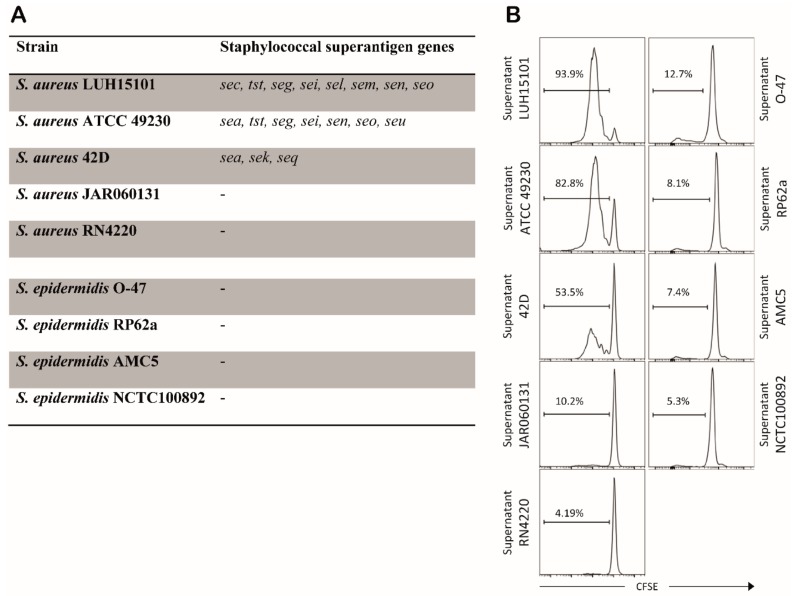
*S. aureus* strains producing SAgs induce high DC-mediated T cell proliferation. (**A**) presence of SAg genes in the different staphylococcal strains. None of the tested strains was positive for *seb*, *sed*, *see*, *seh*, *sej*, *sep* and *ser* gene. (**B**) The CFSE profiles of proliferated allogeneic T cells upon co-culture with DCs and cell-free supernatants of *S. aureus* or *S. epidermidis* strains (^10^log fluorescence intensity). Data from one representative experiment out of three experiments with cells of different donors. The percentage of proliferated cells is indicated.

**Table 1 microorganisms-08-00019-t001:** Primers.

Target Genes	Forward Primer (5’–3’)	Reverse Primer (5’–3’)
*capB*	GCACGAATGCTCTATTGG	CTTCATCTACACCAACTGC
*capC*	GCAGGGTTAGTCGTTCCAG	CGTCAGTATCATGGCAGC
*capA*	GCATTCGTATCACCTATTTAG	CACCAGCATTCGCTAACGC
*capD*	GCGGTTGATGCGGCAATAG	CAGATGTTGTTCATTACTAGGC
*sea*	GAAAAAAGTCTGAATTGCAGGGAACA	CAAATAAATCGTAATTAACCGAAGGTTC
*seh*	CAATCACATCATATGCGAAAGCAG	CATCTACCCAAACATTAGCACC
*sec*	CTTGTATGTATGGAGGAATAACAAAACATG	CATATCATACCAAAAAGTATTGCCGT
*tst*	TTCACTATTTGTAAAAGTGTCAGACCCACT	TACTAATGAATTTTTTTATCGTAAGCCCTT
*sed*	GAATTAAGTAGTACCGCGCTAAATAATATG	GCTGTATTTTTCCTCCGAGAGT
*sek*	ATGCCAGCGCTCAAGGC	AGATTCATTTGAAAATTGTAGTTGATTAGCT
*see*	CAAAGAAATGCTTTAAGCAATCTTAGGC	CACCTTACCGCCAAAGCTG
*seb*	ATTCTATTAAGGACACTAAGTTAGGGA	ATCCCGTTTCATAAGGCGAGT
*sem*	CTATTAATCTTTGGGTTAATGGAGAAC	TTCAGTTTCGACAGTTTTGTTGTCAT
*sel*	GCGATGTAGGTCCAGGAAAC	CATATATAGTACGAGAGTTAGAACCATA
*seo*	AGTTTGTGTAAGAAGTCAAGTGTAGA	ATCTTTAAATTCAGCAGATATTCCATCTAAC
*sen*	CGTGGCAATTAGACGAGTC	GATTGATYTTGATGATTATKAG
*seg*	TCTCCACCTGTTGAAGG	AAGTGATTGTCTATTGTCG
*seq*	ACCTGAAAAGCTTCAAGGA	CGCCAACGTAATTCCAC
*sej*	TCAGAACTGTTGTTCCGCTAG	GAATTTTACCAYCAAAGGTAC
*sei*	CTYGAATTTTCAACMGGTAC	AGGCAGTCCATCTCCTG
*ser*	AGCGGTAATAGCAGAAAATG	TCTTGTACCGTAACCGTTTT
*seu*	AATGGCTCTAAAATTGATGG	ATTTGATTTCCATCATGCTC
*sep*	GAATTGCAGGGAACTGCT	GGCGGTGTCTTTTGAAC

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
