# Peer review of "Dendritic Cells Internalize Staphylococcus aureus More Efficiently than Staphylococcus epidermidis, but Do Not Differ in Induction of Antigen-Specific T Cell Proliferation"

_microorganisms, 2019, doi:10.3390/microorganisms8010019_

Round 1

Reviewer 1 Report

In the manuscript, Balraadjsing et al compared the ability of S. aureus and S. epidermidis to influence dendritic cells (DC) activation and subsequent antigen-specific CD4+ T-cell proliferation in in vitro DC and T cells co-cultures. The authors used human DC derived from peripheral blood monocytes and incubated w/ either S. aureus or S. epidermidis live bacteria or bacterial cell-free supernatants.

Overall, although the manuscript addresses an important area, and better understanding of the interaction between immune cells and these bacteria could inform the development of new therapeutic approaches/ vaccines, the paper lacks novelty and depth. A previous study (Laborel-Preneron et al, PLoS One. 2015; 10(10): e0141067 addressed how S. aureus and S. epidermidis interfered with the maturation /activation of human DC and the downstream consequences of this for CD4+ T cell activation. The present study makes several assumptions (e.g. the secretion of superantigen proteins that are not measured; the absence of superantigens that are not actually tested) and there are no attempts to delve in to the actual mechanisms underlying the differences in T cell proliferation induced by DC infected w/ different S. aureus strains.

Main concerns:

Although data in Figure 3 would indicate that killing of aureus is faster than killing of S. epidermitis, the differential degradation of S. aureus (which is notoriously resistant to intracellular degradation) is not factored in. Since data in Figure 2 shows % of GFP in DC, S. aureus could still be dead, but not completely degraded, which could account for the increased amounts of GFP in S. aureus-infected DC. It would be more informative to compare the % of GFP-expressing DCs in the presence of drugs that could inhibit intracellular bacteria degradation.

Although the authors mention the study by Laborel-Preneron et al, 2015 (ref#5 in the manuscript) they do not address or even discuss the differences in the results between both studies.

The authors assume that the superantigen tested by PCR are actually expressed and secreted in the bacterial culture supernatants, which is not directly tested.

There’s also the assumption that other superantigens, not tested by PCR or otherwise, are not expressed in. the different aureus strains. Although the authors do acknowledge this limitation in the discussion, that does not suffice to improve the manuscript quality. At minimum the authors should make effort to test if supernatants from bacterial cultures that do have superantigens could reverse the effect of supernatants from strain that supposedly lack superantigens.

Minor points:

Figure 4 needs non-stimulated DC controls

Labelling in the graphs shown in 5A-B is confusing: it should be % of divided cells

Author Response

Subject: Manuscript microorganisms-620937

Dear Sir/ Madam,

We thank the editor and reviewers for the review of our manuscript “Dendritic cells internalize Staphylococcus aureus more efficiently than Staphylococcus epidermidis, but do not differ in induction of antigen-specific T-cell proliferation”, authored by P.P. Balraadjsing et al..

We have improved our manuscript based on the reviewers comments and hope that the revised version will be accepted for publication in the special issue “Biology and Pathogenesis of Staphylococcus infection” of Microorganisms. Please find attached our point-by-point replies to the comments of Reviewer 1.

Yours sincerely,

Dr. S.A.J. Zaat

Reviewer 2 Report

The authors report here that the S. aureus is internalised better than S. epidermidis into dendritic cells, but that both species were similar in their ability to induce an antigen-specific T cell proliferation. They suggest that the presence of SAgs may explain the differences in T cell proliferation seen between certain strains of S. aureus and S. epidermidis.

While the conclusions regarding the T cell proliferation and SAgs is justified, some of the conclusions made regarding the intracellular killing of strains need clarification.

Major points

Authors have shown only 15-20% cells internalise S. epi vs 60-70% for S. aureus. By bacterial counts however, the S. epi numbers appear to be only 10-fold less than S. aureus; there are still 10bacteria obtained from cells infected after at 4h. Does this mean that S. epi is replicating more in the cells it enters? The confocal images do not support this. The authors need to comment on this discrepancy. The authors report that the different strains of S. aureus are killed over time in dendritic cells. This is based only on the total bacterial counts. For S. aureus, it has been shown that bacteria replicate and then can lyse the cells and escape causing cytotoxicity (Flannagan et al,2016; Korea et al,2014; Rolin et al 2017). This might explain the decrease in CFU seen with S. aureus strains. Performing cytotoxicity assays on infected cells, flow cytometry at later (24, 48) time points (using the gentamycin protection assay) to see how many total DCs remain relative to earlier times would clarify this. Doing a confocal microscopy analysis at later time points to demonstrate % infected cells would further confirm this. For T cell proliferation experiments, it is not clear if the 48h infected DCs were incubated further with T-cells in presence of antibiotic or not. This would affect the presence of extracellular bacteria (lysing out from infected cells) in the coculture. This needs to be clarified. Again, it is unclear how many infected DC’s were present in these experiments, which as mentioned in 2 may differ especially if the bacteria are lysing out of cells.

Minor points:

Line 67- ‘secretomes induce’- should be ‘induced by extracellular proteins’ or similar

Line 75-77- unclear-please reword

Fig 2 B- the image panels need to be labelled with the species being stained for clarity.

Fig 2C, Figure 4 the X axis should have the strain names- hard to read the vertical columns.

Line 194 and Fig 1: Would be good to clarify that the expression of DC maturation markers was measured by flow cytometry. How long were these were stimulated with bacteria for? Would help the reader if this detail is included in the Legend and Results text.

Why was 17h chosen for the CLIP measurements?

Author Response

Subject: Manuscript microorganisms-620937

Dear Sir/ Madam,

We thank the editor and reviewers for the review of our manuscript “Dendritic cells internalize Staphylococcus aureus more efficiently than Staphylococcus epidermidis, but do not differ in induction of antigen-specific T-cell proliferation”, authored by P.P. Balraadjsing et al..

We have improved our manuscript based on the reviewers comments and hope that the revised version will be accepted for publication in the special issue “Biology and Pathogenesis of Staphylococcus infection” of Microorganisms. Please find attached our point-by-point replies to the comments of Reviewer 2.

Yours sincerely,

Dr. S.A.J. Zaat

Round 2

Reviewer 1 Report

Overall my comments were satisfactorily addressed.

Reviewer 2 Report

The authors have included sufficient new data and clarified concerns raised.